First steps of bipedality in hominids: evidence from the atelid and proconsulid pelvis

Machnicki Allison L. 1
Spurlock Linda B. 2
Strier Karen B. 3
Reno Philip L. 1 plr16@psu.edu
Lovejoy C. Owen 2 olovejoy@aol.com
1 Department of Anthropology, Pennsylvania State University , University Park, PA , United States
2 Department of Anthropology, Kent State University , Kent, OH , United States
3 Department of Anthropology, University of Wisconsin-Madison , Madison, WI , United States
Jungers William
Electronic publication date: 2016 Jan 4
Publication date: 2016
Volume: 4
Electronic Location ID: e1521
Received 2015 Jul 14; Accepted 2015 Nov 29
Copyright: ©2016 Machnicki et al.
Copyright year: 2016
Copyright holder: Machnicki et al.
License: This is an open access article distributed under the terms of the Creative Commons Attribution License, which permits unrestricted use, distribution, reproduction and adaptation in any medium and for any purpose provided that it is properly attributed. For attribution, the original author(s), title, publication source (PeerJ) and either DOI or URL of the article must be cited.
License URL: https://creativecommons.org/licenses/by/4.0/

Keywords: Lordosis, Ardipithecus, Muriqui, Bipedalism, Sacrum, Locomotion, Australopithecus, Proconsul

Funding: National Science Foundation DGE1255832 NSF BCS-0921013 Vilas Research Professors Fund of the University of Wisconsin-Madison CAPES/BRASIL Special Visiting Researcher Program—PVE This material is based upon work supported by the National Science Foundation under Grant No. DGE1255832. Any opinions, findings, and conclusions or recommendations expressed in this material are those of the author(s) and do not necessarily reflect the views of the National Science Foundation. The field study has been supported by a variety of sources to KBS including NSF BCS-0921013, the Vilas Research Professors Fund of the University of Wisconsin-Madison, and CAPES/BRASIL Special Visiting Researcher Program—PVE. The funders had no role in study design, data collection and analysis, decision to publish, or preparation of the manuscript.

==============================
Upright walking absent a bent-hip-bent-knee gait requires lumbar lordosis, a ubiquitous feature in all hominids for which it can be observed. Its first appearance is therefore a central problem in human evolution. Atelids, which use the tail during suspension, exhibit demonstrable lordosis and can achieve full extension of their hind limbs during terrestrial upright stance. Although obviously homoplastic with hominids, the pelvic mechanisms facilitating lordosis appear largely similar in both taxa with respect to abbreviation of upper iliac height coupled with broad sacral alae. Both provide spatial separation of the most caudal lumbar(s) from the iliac blades. A broad sacrum is therefore a likely facet of earliest hominid bipedality. All tailed monkeys have broad alae. By contrast all extant apes have very narrow sacra, which promote “trapping” of their most caudal lumbars to achieve lower trunk rigidity during suspension. The alae in the tailless proconsul Ekembo nyanzae appear to have been quite broad, a character state that may have been primitive in Miocene hominoids not yet adapted to suspension and, by extension, exaptive for earliest bipedality in the hominid/panid last common ancestor. This hypothesis receives strong support from other anatomical systems preserved in Ardipithecus ramidus.

Introduction

Bipedal walking is arguably the most distinctive character of all known hominids.1While the subject of past debate (Stern, 2000), there is general consensus that striding bipedality was maturely developed in all species of Australopithecus as evidenced by a short broad pelvis, adducted great toe, strong bicondylar angle, and a lumbar spine containing 6 post-transitional vertebrae (“functional lumbar” based on zygapophyseal orientation) (Haeusler, Martelli & Boeni, 2002; Kimbel & Delezene, 2009; Lovejoy, 2005a; Lovejoy, 2005b; Lovejoy, 2007; Robinson, 1972; Sanders, 1998; Ward, 2003; Ward, 2013; Williams & Russo, 2015). Bipedal adaptations also are advanced in the earlier hominid, Ardipithecus ramidus, given its short and laterally flared iliac crest and inferred large interauricular distance, and despite its abducent great toe, lateral digit morphology indicating mid-foot rigidity and doming of the metatarsal heads consistent with forceful toe-off (Lovejoy et al., 2009a; White et al., 2009b). Similar adaptations for bipedality also may have been present in Orrorin tugenensis based on proximal femoral morphology (Richmond & Jungers, 2008).

Africa is by far the most likely locus of the adoption of bipedality because the three most closely related clades (panids, gorillids, and hominids) are all also African (Prado-Martinez et al., 2013; Suwa et al., 2007). Environmental reconstructions for Ardipithecus show that it occupied a partially forested and/or closed woodland habitat (Pickford & Senut, 2001; Vignaud et al., 2002; White et al., 2009a). Given environmental trends of the Mio-Pliocene, it is reasonable to presume that the environmental setting occupied by the last common ancestor (LCA) was at least as woodland and that bipedality was not simply a response to occupation of more open habitats. Instead, as evidenced by the simultaneous loss of the sectorial canine complex, the earliest bipedality is probably an element of a broad adaptive shift in social structure (Lovejoy, 1981; Lovejoy, 2009; Suwa et al., 2009; White et al., 2009b).

Characterizations of the initial morphological transition of “how” an ancestral quadruped first became adapted to bipedality have gone largely unexplored. While bipedal walking is achievable for short bouts in apes, such efforts involve a bent-hip-bent-knee gait that will be fatiguing in large bodied animals and would presumably have discouraged its habitual adoption (Biewener, 1989; Crompton et al., 1998). Instead, in all hominids for which the lumbar spine and/or pelvis are available for examination, separation of the most caudal lumbar from the iliac wings permits lordosis necessary for complete hind limb extension. This reflects two major morphological shifts: (1) reduction of iliac height relative to the lumbosacral junction, and (2) relatively broad sacral alae. Both help eliminate physical contact between the transverse processes of the most caudal lumbar vertebrae and the iliac blades, and these two features characterize all known hominids. What is currently unknown is how and when either appeared in hominid phylogeny.

Pelvic anatomy in Ardipithecus

There is mounting evidence that the LCA of panids and hominids lacked most of the specialized (derived) characters associable with suspension and vertical climbing seen in extant African apes [e.g., narrowed sacrum, elongated iliac isthmus, modification of femoral insertion characters of the hip musculature, reduction in the number of lumbar vertebrae to an average of 3.5, elongated sacra (by conversion of somite fates from lumbar to sacral) and elongated thorax (by conversion of somite fates to thoracic—especially advanced in the bonobo)] (Almeçija et al., 2013; Crompton, Vereecke & Thorpe, 2008; Kivell & Schmitt, 2009; Lovejoy & McCollum, 2010; McCollum et al., 2010; Reno, 2014; Thorpe, Holder & Crompton, 2007; Ward, 2015). It is unlikely that these characters were present in the LCA, since the metacarpus, carpus, limb proportions, foot, femur, humerus and ulna in ARA-VP-6/500 (“Ardi”) exhibit morphologies that lack definitive modifications for suspension (Almeçija, Smaers & Jungers, 2015; Lovejoy et al., 2009a; Lovejoy et al., 2009b). They are instead more consistent, based on the considerably likely substantial body mass of Ardipithecus, with deliberate quadrumanual climbing and bridging accompanied by ulnar withdrawal and posterolateral shoulder relocation with attendant invagination of the vertebral column (see below) (Lovejoy et al., 2009c; White et al., 2015). These were accompanied by substantial lateral enlargement of the iliac blade and reduction of the retroauricular region of the os coxa in Ar. ramidus indicating spinal invagination into the thorax (Lovejoy & McCollum, 2010; Lovejoy et al., 2009d).

Exceptions to the generalized primitive state in Ar. ramidus are its “upper” pelvis and lateral foot, which are consistent with a functional hip stabilization mechanism and rigid toe-off during upright walking. Its ilium is similar to those of later hominids and includes an anterior inferior iliac spine likely developed by a separate ossification center, a hominid apomorphy. Unfortunately, the specimen lacks both its lumbar column and a sufficient amount of its sacrum to permit direct demonstration of its capability for lordosis (Lovejoy et al., 2009a; Lovejoy et al., 2009b; Lovejoy et al., 2009c; Lovejoy et al., 2009d). This presents an interpretive conundrum. How can we then deduce how lordosis was likely achieved in the earliest phases of upright walking in the hominid clade?

The lumbosacral and pelvic anatomy of the LCA

The pelvis of the middle Miocene proconsulid, Ekembo nyanzae (McNulty et al., 2015), was generally similar to those of Old World monkeys (Ward, 1991; Ward, 1993; Ward et al., 1993). Its ilia were superoinferiorly long with a substantial gap between the sacral promontory and pubic symphysis, and the iliac fossa was quite narrow. The os coxa exhibits a relatively massive retroauricular portion (i.e., the iliac tuberosity) (Ward, 1991). It is now clear that proconsulids and other Miocene hominoids such as Nacholapithecus lacked tails (Nakatsukasa et al., 2003a; Nakatsukasa et al., 2003b; Nakatsukasa et al., 2004; Ward, Walker & Teaford, 1991), even though most were still largely above branch quadrupeds (Beard, Teaford & Walker, 1986; Morbeck, 1975; Napier & Davis, 1959; Ward, 1993; Ward, 2015). Tail loss was likely a hominoid synapomorphy by at least 17.9 mya (Nakatsukasa et al., 2004). The lumbar column was long, probably numbering six or seven ribless vertebrae (McCollum et al., 2010; Ward, 1991).

A dramatic shift in pelvic proportions is present in Ar. ramidus. The relative sizes of its pre- and retro-auricular portions are almost fully modern and very unlike those of Ekembo having been transformed by substantial vertebral column invagination and very likely migration of the lumbar transverse processes from mid-centrum to pedicle (Lovejoy et al., 2009c; White et al., 2009b). A pelvic fragment of the Miocene taxon Pierolapithecus catalaunicus (Hammond et al., 2013) and the os coxa of the late Miocene Oreopithecus bambolii have been described (Harrison, 1986; Harrison, 1991; Rook et al., 1999; Straus, 1963; Wood & Harrison, 2011), although the latter’s poor state of preservation appears not to have been fully appreciated (White et al., 2015). Arguments that its lumbar vertebral bodies show wedging to facilitate lordosis (Kohler & Moya-Sola, 1997) have been rigorously rejected (Russo & Shapiro, 2013). Moreover, the os coxa’s extreme compression during fossilization puts little confidence in claims that it exhibited a “true” anterior inferior iliac spine (White et al., 2015), as its general form is grossly inconsistent with origin by a secondary apophysis which characterizes the structure in all known hominids (contra Harrison, 1991; Harrison & Rook, 1997; Straus, 1963). Nevertheless, these Miocene fossils do provide a basis for reconstructing the LCA’s general pelvic form.

The iliac isthmus in late Miocene pelves must have still been superoinferiorly long with a substantial “promontory-symphysis vertical gap,” but likely with some lateral expansion (i.e., “flaring”) of the iliac fossa (Hammond et al., 2013). P. catalaunicus suggests some modifications of the limbs, thorax and pelvis for more competent arboreal clambering and deliberate climbing (including ulnar withdrawal and a more elliptical thorax than monkeys and proconsulids), but not to the degree seen in Ar. ramidus (Moya-Sola et al., 2004). This is especially true of the position of the lumbar transverse processes, whose origins were not yet fully pedicular as in hylobatids and other Miocene taxa such as Hispanopithecus and Morotopithecus (MacLatchy, 2004; Moya-Sola & Kohler, 1996). In the present paper we distinguish “deliberate climbing” as predominantly above branch cautious based on the relatively large body mass of several Miocene specimens that would render the leaping and acrobatic behavior seen in smaller primates hazardous because of substrate reactive elasticity. We distinguish “vertical climbing” (sensu stricto, as opposed to generally climbing upward) as that form of ascension of large trunks performed specifically by living African apes (see DeSilva, 2009; Fleagle et al., 1981 for illustration). These are not necessarily mutually exclusive within a given primate’s locomotor repertoire, however they do impose different functional demands and likely selective pressures on the skeleton depending on their relative frequency.

This pelvic form of Miocene apes such as Ekembo, when combined with a relatively long lumbar column, provides a basis for two key questions concerning the acquisition of lordosis: (1) how permissive were the vertebral column and pelvis of the LCA for lordosis, and (2) which of the two morphological shifts observed in Ardipithecus (reduction of iliac height and/or expansion of sacral breadth) was more likely to have occurred first in the evolution of earliest hominid bipedality?

Atelids as a model for the relationships between the pelvis and lumbar column

New World atelids (Alouatta, Ateles, Brachyteles, and Lagothrix) are unique because they frequently engage in caudal suspension (Cant, 1986; Iurck et al., 2013; Johnson & Shapiro, 1998; Lemelin, 1995; Mittermeier & Fleagle, 1976), which often causes their lower spines to enter into substantial sagittal recurvature similar to lordosis in hominids (Fig. 1A). The muriqui (Brachyteles) and spider monkey (Ateles) are particularly relevant as they have also undergone lumbar column reduction resulting in approximately 6 post-transitional vertebrae similar to early hominids (Williams, 2011). Examination of their pelvic and sacral morphology therefore may provide indirect evidence of the potential pathway toward lordosis in earliest hominids.

Figure 1 Bipedal posture in muriquis.

Lumbar lordosis facilitates an erect trunk and fully extended hind limb in northern muriquis. (A) Female in partial caudal suspension; note marked lordosis. (B) Male; note complete extension of the right lower limb. (C) Male; note complete extension of both hind limbs and fully erect trunk and that the tail is not being used for support. (D) Female with infant preparing to climb; note that the left hind limb is in extension in combination with an entirely vertical trunk. (E) Adult male standing without brachial or caudal support; note the extended back. The bipedal postures shown in (B)–(C) and (E) were adopted during brief resting bouts, while that in (D) was adopted during a transition from terrestrial to arboreal substrate. See Videos S1 and S2 for context of typical bipedal behaviors. Photo credits as follows: A, Daniel Ferraz; B, D & E, Fernanda P. Tabacow; C, Marina Schultz de Cristo.

In order to address this, we have reviewed lumbar and pelvic form and function in these primates. First, we predict that the ability to generate lordosis should facilitate the attainment of more erect hind limb postures. To address this, we report observations of terrestrial behavior of northern muriquis (Brachyteles hypoxanthus) and utilize opportunistic observations of bipedal locomotion in Ateles. Second, we hypothesize that lordosis will be accompanied by features associated with caudal lumbar emancipation. As such, we investigate iliac height, lower lumbar ligamentous support, and sacral width in relevant extant and fossil anthropoids. Finally, we provide a possible scenario for the role of pelvic and sacral form in the evolution of bipedality.

Methods

Brachyteles bipedality

Observations of northern muriquis (B. hypoxanthus) were conducted at the Reserva Particular de Patrimônio Natural-Feliciano Miguel Abdala in Caratinga, Minas Gerais, Brazil. In recent years, as a consequence of substantial local demographic change, we have noted an increase in the frequency and duration of bouts of terrestrial behavior in one population of unprovisioned northern muriquis living at high density (Mourthé et al., 2007; Tabacow, Mendes & Strier, 2009). These enable terrestrial behavior to be monitored in a naturalistic setting. We photographed and video recorded bouts of terrestrial behavior to determine the general nature of their bipedal posture (Fig. 1 and Videos S1 & S2). Methods were approved by the University of Wisconsin Animal Care Committee (protocol L00104 through April 2011; then a Wildlife Waiver). The Brazilian government, CNPq, and the administration of the field site provided permission.

Given the likely kinematic variability in bipedal locomotion of non-human primates, we also surveyed video resources for further examples of bipedal behavior of atelids (youtube.com and arkive.com using a combination the search terms “spider monkey,” “Ateles,” “muriqui,” “woolly spider monkey,” or “Brachyteles” and “bipedal” or “walking”). As these involved opportunistic observations the descriptions we have provided are necessarily qualitative.

Iliac and sacral anatomy

In order to compare the ligamentous anatomy of primates (especially atelids and Old World monkeys) with free (potentially lordotic) versus constrained (restrictive of significant lordosis) last lumbar vertebrae, we conducted detailed dissections of spider (Ateles sp.), muriqui (Brachyteles sp.), howler (Allouata sp.), and langur (Presbytis sp.) monkeys as well as a single gibbon (Hylobates sp). The first three taxa allow a comparison of the shortened and longer backed atelids, and the langur was chosen because it should represent the more generalized catarrhine condition. The gibbon provides a comparison to hominoid lumbosacral and pelvic anatomy absent the extreme specialization observed in the large-bodied great apes. These specimens are housed in the collections of the Cleveland Museum of Natural History, Kent State University, Northeastern Ohio Medical University, and Case Western Reserve University.

We also collected metric and nonmetric data from the pelves of 150 skeletonized specimens in the Cleveland Museum of Natural History and Harvard Museum of Comparative Zoology (Table 1 and Table S1). Unfortunately, we did not obtain a suitable sample of measurable muriqui specimens, but we did obtain metrics from each of the three other atelid genera. We paid particular attention to the position of the sacral promontory in relation to the iliac crest and ischiopubic ramus (Fig. 2). From these data we calculated the relative iliac height above the sacral promontory within the pelvis (see Fig. 3 and its legend). For size normalization we used acetabular diameter (Plavcan, Hammond & Ward, 2014) to calculate the Iliac Height Ratio. To determine the relationship between alar breadth and centrum breadth, we performed an analysis of covariance (ANCOVA) between genus means of log alar breadth and log centrum breadth with a binary categorical variable (monkey versus ape) as a covariate. To account for potential autocorrelation in related taxa, we used a phylogenetic generalized least squares (pGLS) regression in the ‘caper’ package in R (Orne et al., 2011; Team , 2012). We obtained the phylogenetic tree and branch lengths from the 10kTrees Project (Fig. S1) (Arnold, Matthews & Nunn, 2010).

Table 1 Comparative sample of pelvic metrics.

Species	N	Species	N	
Atelids	24	Old World monkeys	27	
Alouatta palliata	6	Cercopithecus torquatus	2	
Alouatta guariba	3	Cercopithecus mona	1	
Alouatta caraya	1	Chlorocebus aethiops	6	
Alouatta seniculus	1	Colobus guereza	4	
Aloutta sp.	1	Macaca fascicularis	1	
Ateles geoffroyi	8	Macaca mulatta	1	
Ateles sp.	1	Macaca silenus	1	
Lagothrix lagotricha	3	Papio hamadryas	3	
		Presbytis cristata	1	
Other New World monkeys	32	Presbytis rubicunda	2	
Aotus vociferans	1	Pygathrix sp.	1	
Calllicebus sp.	4	Semnopithecus entellus	1	
Callithrix jacchus	2	Theropithecus gelada	2	
Cebuella pygmaea	2	Trachypithecus pileatus	1	
Cebus albifrons	2			
Cebus capucinus	4	Hominoids	67	
Cebus apella	4	Homo sapiens	20	
Cebus sp.	1	Pan troglodytes	15	
Chiropotes santanas	1	Gorilla gorilla	15	
Leontopithecus rosalia	4	Pongo pygmaeus	7	
Saguinus geoffroyi	1	Hylobates lar	10	
Saguinus midas	1			
Saguinus oedipus	1	Total	150	
Saimiri sciureus	4			

Figure 2 Comparison of iliac height and lumbar entrapment.

(A) Comparison of the relationships between the most caudal lumbar in a langur (Presbytis, left) and howler monkey (Alouatta, right). The deep iliac “well” in which the langur’s L7 is positioned limits its potential motion through ligamentous attachments (see Fig. 5), whereas the howler monkey’s L5 is more mobile by virtue of the reduced height of its ilia. Metric definitions are indicated. Dashed line is for reference; solid lines indicate measured distances. Iliac height = A; Acetabular breadth (not visible) = B; Centrum breadth = C; Total sacral breadth = D; Alar breadth = (D − C)/2. (B) Pelvis of an adult male muriqui (Brachyteles) demonstrating the low iliac height and free caudal lumbar vertebrae typical of atelids. This individual is from the same study group as those depicted in Fig. 1. Specimen housed at the Museu Nacional Rio de Janeiro (National Museum of Brazil). Photo credit: Sérgio L. Mendes. (C) In this chimpanzee (P. troglodytes), vertebral motion is restricted by the direction of their transverse processes and ilia. Note the narrow inter-iliac distance as marked. Arrows indicate articulation between lumbar transverse processes and iliac crest. It is important to note that the langur condition (A, left) is not equivalent to that in extant African apes.

Figure 3 Box plot illustrating relative iliac height across anthropoids.

Iliac Height Ratio = (Iliac height X 100)/Acetabular breadth. Analysis of Variance (ANOVA) demonstrates highly significant difference between means whether atelids are grouped as a family or as individual species. Significance values indicated from Bonferroni posthoc pairwise test between combined atelids and other groups (∗∗, p < 0.01; ∗∗∗, p < 0.001). Atelids that engage in caudal suspension most often have the lowest iliac height except for humans. The three New World monkey lower outliers are specimens of Chiropotes santanus, Aotus vociferans, and Callicebus sp.

Proconsulid sacral reconstruction

The partial skeleton of Ekembo nyanzae (formerly Proconsul) from Mfangano Island, Kenya, includes a portion of the first sacral body (S-1:KNM-MW 13142-M) (Ward et al., 1993). We used a Kenya National Museum cast as the “core” of our reconstruction of the sacrum. The specimen’s inferior portion includes the rim of the first sacral foramen on the left side, thus indicating the craniocaudal height of the S-1 body, and its posterior portion indicates overall thickness of S-1 just lateral to the left articular process (see Fig. 2 in Ward et al. 1993). The superior surface of the S-1 centrum is essentially intact with only minor abrasion. We used “extra firm” oil-based modeling clay to build the left ala and the (missing) articular facets of S1, incorporating but not obscuring the fossil’s S-1 fragment. Elements 2-5 were added as well to create a reasonable facsimile of the likely entire structure of the original specimen, although these details have no bearing on the role of our reconstruction for the current report. Most importantly, for the auricular portion of the sacrum, clay was molded to exactly conform to the well-preserved auricular surface of the nearly complete os coxa of KNM-MW-13142-D (also a Kenya National Museum cast). This molded surface was then mated to the S-1 cast absent the addition of any material not needed merely to successfully fuse the clay auricular mold to the left side of the specimen’s S-1 plaster body as preserved in the cast. Ward et al. (1993) describe the left ala as being “preserved adjacent to the first sacral body, but [with] its ventral edges…eroded away” (p. 84). Our reconstruction thus minimized the effects of this erosion and produced essentially a 3-D version of Ward et al.’s drawing of the likely pelvic structure of the specimen (their Fig. 14), although their version appears to indicate the addition of more material than was actually required for our version (Fig. S2). Nevertheless, the latter produced a slightly broader ala than indicated in their scaled drawing (see below). Reconstruction was completed by mirror imaging the (missing) opposite side. The clay model was then molded and cast in plaster. Dimensions of the original specimen are available from Ward et al. (1993:85). They report that the “articular surface of the first sacral body is elliptical in outline, 19.6 dorsoventrally and 30.0 mediolaterally.”

Results

Lordosis and upright posture in atelids

Atelids have previously been observed during bouts of terrestrial behavior only rarely (Campbell et al., 2005; Dib, Oliva & Strier, 1997; Mourthé et al., 2007). Systematic analyses of substrate use in Brachyteles showed that by 2005 muriquis of all age-sex classes in our study group occasionally engaged in terrestriality, and by 2007 adult males were spending 1.5% of their time on the ground, nearly a 50% increase from the 0.8% of time they spent on the ground in 2006 (Tabacow, Mendes & Strier, 2009). Their terrestrial activities had also diversified to include both essential ones (e.g., drinking, traveling across gaps in the forest, and feeding) and nonessential ones (e.g., socializing, including mating, resting, and traveling in areas where arboreal alternatives were available). Increasing terrestriality has continued to be present in this group and in the other three muriqui groups in our study population through July 2015 (K Strier, 2015, unpublished data). In 2011, with the use of strategically-deployed motion-sensitive camera traps, we were able to demonstrate for the first time that their terrestriality occurs in the absence of observers. Video images (Videos S1 and S2) show adult male muriquis descending to the ground to feed on fallen fruits in an open area within the forest, engaging in reassuring social contact as they move through the area with a combination of quadrupedal and bipedal postures. Thus, despite the overall rarity of terrestriality in Brachyteles, examination of posture in these naturalistic bouts makes it clear that individuals can readily achieve both a fully erect trunk and a near complete extension of the lower limb. Examples of these postures are shown in Figs. 1B, 1C and 1D. These observations indicate that the conformation of muriqui lumbar, pelvic and sacral anatomy is sufficient to produce fully extended hind limb postures, and that such behaviors are not dependent on habituation or training in captive animals. In particular, Figs. 1B and 1C indicate that hind limb extension can be achieved without the substantial lateral rotation of the limb observed in orangutans (see below) (Stern & Larson, 1993; Tuttle, 1979).

We also searched for other instances of videos depicting bipedal posture and locomotion in atelids to confirm the capacity of bipedal posture and locomotion in other species. As expected these tended to be highly variable using a variety of postures given their primary use of quadrupedal locomotion. However, opportunistic observations of spider monkeys (Ateles sp.) demonstrate that near or actually fully extended hip and knee postures can be achieved during bouts of bipedal locomotion in this species as well (Fig. 4 and Video S3).

Figure 4 Bipedal posture in a spider monkey.

Screen captures of video demonstrating bipedal posture and locomotion. (A) Posterior lateral view demonstrating that the spider monkey attains erect hip and back during bipedal standing without relying on support. (B) Lateral view illustrating that hip extension approximates 160°. Images from BigLivigVideos, (2012); full video can be seen as Video S3.

Pelvic height reduction

To test the hypothesis that lordosis and capacity for erect posture are associated with caudal lumbar emancipation, we compared relative iliac height above the sacral promontory among various anthropoids. When atelids, such as a howler monkey and a muriqui, are compared to other anthropoids, such as a langur and a chimpanzee (Fig. 2), it is visually apparent that the most caudal lumbar is not restricted by contact or bilateral ligamentous attachment to the dorsal portions of their shortened iliac blades. Metrically this can be confirmed by an analysis of the Iliac Height Ratio (Fig. 3). Our metric successfully isolates atelids from other monkey and ape groups. Of note is the similarly reduced pelvic height in humans, which may in fact be accentuated by a relatively large acetabulum diameter (Jungers, 1988). These results demonstrate that atelids, including the less suspensory Aloutta, have reduced iliac height resulting in relatively unencumbered caudal lumbar vertebrae.

Ligamentous support

We further explored if mechanisms associated with caudal lumbar mobility extended to also include soft tissues. Either direct articulation of the transverse processes or bilateral ligamentous attachment would limit mobility in vertebrae located directly between the iliac crests. Each of the atelids and the gibbon had reduced lumbar columns relative to those of the langur and the presumed catarrhine ancestral condition (McCollum et al., 2010; Pilbeam, 2004; Schultz & Straus, 1945). All of the dissected atelids exhibited a free last lumbar vertebra, while the most caudal lumbar in the langur and gibbon were instead located between the more dorsally extended ilia confirming previous observations (Figs. 2 and 3). Dissection revealed substantially less ligamentous restriction in the atelids than in the gibbon and langur, but ligamentous tissue was generally denser and more elaborate in the langur than in the gibbon (Fig. 5). Presumably this is related to the retention of a massive erector spinae in the cercopithecoids such as the langur (Benton, 1967), and loss of the tail (with some partial invagination of the spine) in the gibbon. It appears that any lordosis in the langur is accomplished mostly by the superoinferior length and number of its lumbar and post-transitional thoracic vertebrae (along with some presumed differential disc compression) as observed in macaque monkeys trained to habitually walk bipedally (Preuschoft, Hayama & Günther, 1988).

Figure 5 Iliolumbar (1), intertransverse (2), and iliosacral (3) ligament anatomy observed during primate dissections.

Superficial ligaments are shown on the left, and deeper tissues shown on the right. Lumbar numbers are based on rib count, as articular facet orientation could not be observed without further destructive dissection. (A) Spider monkey (Ateles) [3 lumbars] and (B) spider monkey [4 lumbars]: Iliolumbar ligament spans L2 or L3 and the ilium, and a thin intertransverse ligament spans the transverse processes of L3 or L4, ilium and sacrum. These ligaments likely provide lumbar support, while permitting substantial mobility. (C) Muriqui (Brachyteles) [5 lumbars]: an iliolumbar ligament runs from L3, L4, and L5 to the ilium. A thin intertransverse ligament spans the transverse processes of the lumbar vertebrae. As in the spider monkey, these ligaments likely provide lumbar support, while permitting substantial mobility. (D) Howler monkey (Alouatta) [5 lumbars]: an iliolumbar ligament runs from L4 and L5 to the ilium. This tissue is narrower and covers less surface area on the ilium than the corresponding ligament in the spider monkeys, but serves a similar function. A thin intertransverse ligament spans the transverse processes of the lumbar vertebrae and runs from L5 to the ilium. (E) Gibbon (Hylobates) (5 lumbars): a thick iliolumbar ligament runs horizontally and obliquely between L3, L4, and L5 and the ilium. A thick intertransverse ligament run between each lumbar transverse process. The orientation of the ligamentous fibers is more similar to those of atelids than those of the langur, but the thickness was intermediate between the two. The ligamentous tissue would not have been as restrictive as in the langur. (F) Langur (Presbytis) [7 lumbars]: iliolumbar ligaments run from the 3 most caudal lumbar vertebrae to the ilium. A thick intertransverse ligament spans the transverse processes of the lumbar vertebrae and runs from L7 to the ilium to join both structures to the transverse processes of L6. Since the L7 is positioned deeply between the iliac blades, its motion is highly restricted. The ligamentous tissue of the langur is substantially denser than that of all the other specimens and would have restricted motion more substantially. The iliolumbar ligament is hypothesized to have developed in primates associated with stabilizing the back while upright or lordosing and is possibly formed by collagenation of the fibers of the quadratus lumborum (Luk, Ho & Leong, 1986; Pun, Luk & Leong, 1987). It has not been found in other quadrupedal animals like cats and dogs (Pun, Luk & Leong, 1987).

Sacral alar breadth

Another osteological feature that can impact lumbar mobility is the breadth of the sacrum and its effect on interauricular distance. To determine relative sacral breadth in atelids, we normalized alar breadth by the transverse breadth of the first sacral centrum (as a measure of body size) in anthropoids (Fig. 6). A comparison of the two regression lines representing monkeys and apes is of interest. The sacra of both New World and Old World monkeys, including atelids, are broader than those of suspensory apes. In contrast, Homo and Australopithecus individuals plot well above the combined monkey regression line. This shows that atelids possess a broad sacrum similar to those of other monkeys and that apes alone are distinguished by extreme sacral narrowing.

Figure 6 Relative sacral breadth in anthropoids.

Sacral alar breadth compared to centrum breadth (for definitions see Fig. 2). Data represents genus means for extant taxa. Regression formulas are provided from a phylogenetic Analysis of Covariance (ANCOVA). Slopes do not differ significantly; however, there is a significant difference in elevation (p < 0.0001) between monkeys (long dashes) and apes (short dashes). Two points are plotted for Ekembo (KNM-MW-13142) representing the two alar breadth estimates discussed in the text. Hominids, including Au. afarensis (A.L. 288-1 and KSD-VP-1/1), Au. africanus (Sts-14), and H. erectus (BSN49/P27 Simpson et al., 2008) plot above the monkey regression line. The two non-atelid New World monkeys falling below the monkey regression line are Chiropotes santanas and Callicebus sp.

The early Miocene hominoid sacrum

It is possible that the especially narrow sacrum in apes might be a product of tail loss and not selection for lumbar entrapment. As such, it is of great interest to know the sacral dimensions in KNM-MW-13142, since it exhibits a primitive pelvis and lumbar column after tail loss. However, the specimen’s lateral alar portions are both partially eroded. Ward reported the mediolateral breadth of the first sacral centrum to be 30 mm (see ‘Methods’), and that “the farthest lateral point on the preserved portion of the wing is 25.7 from the midline” (Ward et al., 1993:85, emphasis added). This would be an unrealistic minimum value of 10.7 mm for alar breadth less centrum breadth (25.7–30/2), since a substantial portion of each ala is obviously missing—the fundamental question being “how much?” Ward et al. provided a drawing of their reconstruction of the pelvis (see Fig. 14 in Ward et al., 1993), and based on its scale they appear to have added approximately 7 mm to each ala to complete its pelvic ring.

The left os coxa of KNM-MW-13142 is nearly complete and includes an intact auricular surface and almost the entirety of the lower pelvis, which lacks only a small portion of the pubic symphysis. We physically reconstructed the missing portions and its sacrum to form a realistic true pelvis (Fig. S2, see earlier). Essentially this required adding only sufficient material to each ala (presuming bilateral symmetry) to fully articulate the (missing) sacral auricular surface with the three-dimensional surface of the os coxa—the posterior portion of the latter being substantially involuted. The reconstructed sacrum’s dimensions are in full agreement with those of Ward et al.’s. These would seem to be a reasonable minimum because the strong mediolateral angulation (about 45°) of the specimen’s auricular surface requires the anterior surfaces of its sacral alae to be as least as large as the ones both Ward and we reconstructed. Admittedly these are crude estimates (theirs at 17.7 mm and ours at 18.7 mm), but when plotted in Fig. 6 (see Table S1), the specimen falls above the regression line defined by monkeys and not with that representing the extant African apes.

Such a reconstruction, although seeming quite reasonable, should be considered here mainly as a means of illustrating a hypothesis, which will hopefully be tested by and the recovery of more complete specimens. In any case, these data do indicate that the hominoid sacrum did not undergo substantial reduction in breadth simply due to tail elimination.

Discussion

The potential parallelisms between atelids and hominoids with respect to suspensory locomotion have long been observed (Erickson, 1963; Larson, 1998). However, the lumbar column, thorax, and pelvis of spider, muriqui, woolly, and howler monkeys differ substantially from those of hominoids in many ways, especially since monkeys retain external tails. The presence of a tail is also shared with Old World monkeys, although those of atelids are more massive, highly innervated, and prehensile. As in other monkeys, their iliac isthmus is superoinferiorly long with a substantial gap between the sacral promontory and pubic symphysis. In one crucial respect, the atelid spine resembles that of hominoids. Atelids have undergone a significant degree of spinal invagination which is likely associated with their mediolaterally broad thorax and potentially enlarged prehensile tail. Typical atelid lumbars are shown in Figs. 7 (for muriqui see Fig. S3), and it is quite clear that their transverse processes are much more dorsally located than are those in specimens of cercopithecoids and some early hominoids (e.g., Ekembo) (Kagaya, Ogihara & Nakatsukasa, 2008; Kagaya, Ogihara & Nakatsukasa, 2009). Thus, atelids differ from other monkeys because they have evolved partial invagination similar in degree to that in gibbons and Pierolapithecus, Hispanopithecus and Morotopithecus (Figs. 7) (MacLatchy, 2004; Moya-Sola & Kohler, 1996; Moya-Sola et al., 2004).

Figure 7 Ultimate lumbar vertebrae in atelids and an Old World monkey.

Note that transverse process position in atelids shows partial invagination. (A) Howler monkey (Alouatta, CMNH 1172) L5. (B) Spider monkey (Ateles, CMNH 1237) L5. (C) Woolly monkey (Lagothrix, CMNH 5699) L4. (D) Colobus monkey (Colobus, CMNH 2144) L7. The transverse process location in atelids is similar to that in gibbons and Pierolapithecus (Moya-Sola et al., 2004). In the ‘semi-brachiating’ Colobus, the transverse processes originate from the vertebral body.

Brachyteles individuals achieve bipedal posture with both an extended torso (at the hip) and knee joints that are accompanied by a lordotic curvature of the lower back. Ateles can also attain a similar posture and can walk with a more extended gait than typical of African apes and other monkeys (Fig. 4, Video S3) (Okada, 1985; Stern & Larson, 1993). For example, Okada (1985) reports that a spider monkey attains a maximum hip angle of nearly 160° during bipedal walking and an angle of 140° at toe off. Each of these are approximately 20° greater than observed in a chimpanzee and gibbon (Okada, 1985). Comparisons between studies shows there is some variation between achieved hip angles, particularly in smaller bodied spider monkeys and gibbons as Yamazaki observed gibbons attaining greater limb extension (Yamazaki, 1985). With the caveat that it is difficult to know what motivates individual primates to attain a particular range of extension during short bouts, it is noteworthy that larger bodied chimpanzees do not achieve more extended postures, despite the greater kinematic pressures to do so (Biewener, 1989). Regardless, the observations of Okada (1985) approximately match the levels of extension observed in the spider monkey in Fig. 4. This is despite the fact that atelids retain only 4–5 ribless and approximately 6 functional lumbar vertebrae (Williams, 2011). Full extension of the hind limb is obtainable in other primates during overhead reaching, but the unusual attribute available to these atelids is their additional capacity to also simultaneously extend the hip, a combination that appears lacking in extant African apes (Pontzer, Raichlen & Rodman, 2014; Tuttle, 1979). Orangutans also achieve substantial hip extension despite a short spine and lumbar entrapment; however, these bouts tend to be brief when lacking forelimb support and involve substantial lateral rotation of the hind limb (Stern & Larson, 1993; Thorpe, Holder & Crompton, 2007; Tuttle, 1979). Lordosis in atelids appears to be facilitated by a shorter iliac crest and reduced ligamentous restriction of the caudal lumbar vertebrae, as well as partial lumbar invagination. These, in combination with their broader sacrum, allow the caudal lumbar vertebrae to contribute to lordosis.

It is important to note that relatively low iliac height and spinal invagination also occur in Alouatta as in other atelids (Figs. 3 and 4). In combination with potential parallel evolution of suspensory specializations in Ateles and Brachyteles (Iurck et al., 2013; Jones, 2008), this indicates that these features need not have evolved in the context of forelimb brachiation or suspension. Instead, the ability to enter lordosis in atelids was likely a response to occasional tail-assisted hind limb suspension and diverse forelimb loading postures in an otherwise arboreal quadruped (Cant, 1986). This raises a very interesting question with respect to the origins of upright walking, which requires at least partial lordosis for reasonable success. Since the LCA obviously lacked a prehensile tail, we may ask whether spinal invagination, which was part of the major shift in bauplan that permitted lateralization of the shoulder, was not also a critical exaptation that would eventually facilitate the adoption of upright walking in a descendant of the LCA? Based on transverse process position in some Miocene hominoids such as Pierolapithecus and a similar transverse process location in atelids, we suggest that there is a strong probability that it was. These data enable us to hypothesize the process by which lumbar lordosis could have evolved in the earliest hominids.

African ape sacra have strikingly narrowed alae (Fig. 6). This is very likely an adaptation to vertical climbing and/or suspension, and includes a reduction in lumbar number by conversion of lumbar to thoracic and/or sacral vertebrae and the entrapment of caudal lumbar segments between the ilia. Together these render the ape spine virtually rigid (Lovejoy & McCollum, 2010; McCollum et al., 2010; Schultz, 1969). As a consequence, the African apes cannot easily locate their center of mass over their pedal support and must rely on the classic bent-hip-bent-knee gait during upright walking (Fleagle et al., 1981).

Figure 8 Models depicting the evolution of the lumbar column, sacrum, and ilium in hominoids.

(Top) Early Miocene hominoids such as Ekembo had an uninvaginated spine, long lumbar columns (≥6 vertebrae), wide sacrum, and tall iliac height (sacral promontory/iliac crest distance) similar to other generalized catarrhines. The presence of Eurasian Miocene hominoids with ancestral morphologies (indicated by dashed line) suggests that gibbons and orangutans invaginated their spines (A) and narrowed their sacra (C) in parallel. The orangutan lumbar column length was further shortened to 4 elements (B). Shared morphologies between African apes, humans and Ardipithecus (i.e., reduced retroauricular portion of the pelvis) indicate spinal invagination and broader thorax with dorsally placed scapula were characteristic of the ancestors of the African ape clade. However, the lack of derived suspensory features in arboreal Ardipithecus, suggests that the LCA retained a long lumbar column and wide sacrum. Thus, lumbar reduction (B) and sacral narrowing (C) occurred in parallel among African apes, and iliac height reduction (C) occurred in early hominids (5 transitions) as an initial adaptation for bipedality. In New World monkeys, similar morphologies evolved in parallel with spinal invagination(A) and reduced iliac heights (D) occurring in the common ancestor of atelids, with lumbar column reduction occurring in spider monkeys (B) and muriquis (not shown). (Bottom) An alternative model posits that the African ape and human LCA had already evolved numerous suspensory and vertical climbing specializations including spinal invagination (A), lumbar reduction (B) and sacral narrowing (C). In such a case, lumbar length (B’) and sacral width (C’) would have reversed in early hominids. Both models provide similar numbers of evolutionary transitions. Note that our depiction of these models does not include the additional transitions to 6 post-transitional vertebrae in Australopithecus and a return to 5 in humans nor potential parallel lumbar shortening in chimpanzees and bonobos (McCollum et al., 2010).

Given the importance of lordosis for achieving extended hind limb bipedal posture, it is likely that significant alar breadth was retained in the LCA and was a feature in earliest hominids (Fig. 6). The alternative hypothesis, that alar breadth and lumbar column length were first reduced in mid-Miocene hominoids only to then be re-broadened in bipedal hominids is decidedly more complex than the more modest alternative that the Ardipithecus postcranium presents, i.e., that some late Miocene taxa had undergone modifications for effective clambering and cautious climbing without sacral narrowing. The lack of derived suspensory adaptations in the innominate as well as other parts of the skeleton in early members of the Pongo clade such as Sivapithecus (Madar et al., 2002; Morgan et al., 2015; Pilbeam et al., 1990) suggest that lumbar shortening and sacral narrowing very likely occurred in parallel in Asian apes (Larson, 1998; Ward, 2015) (Figs. 8A). In addition, there is evidence of parallel experimentation with a similar bauplan shift in Morotopithecus, which has sufficiently dorsal placement of the transverse processes to suggest spinal invagination, broadening of the thorax, and lateralization of the scapula similar to their counterparts in atelids and gibbons (MacLatchy, 2004). Within the African ape and human clade, when simply considering character state transitions of lumbar spine length, sacral width and iliac height, homoplasy due to reversal is equivalent to parallel evolution between lineages. If the lumbar column was reduced and the sacrum narrowed prior to the African ape and human LCA, these features would have necessarily been reversed in early hominids while iliac height was reduced (Figs. 8B). Alternatively, lumbar reduction and sacral narrowing could have occurred in parallel between the Gorilla and Pan lineages, and simple iliac height reduction occurred in early hominids (Figs. 8A). Each of these scenarios posits five transitions associated with the evolution of bipedality and suspensory/vertical climbing behaviors (with the caveat that the developmental independence of these characters and the number actual genetic transitions is completely unknown). However, reversal implies fluctuating selective pressures in a single lineage. And Ar. ramidus does not exhibit any of the numerous additional adaptations to suspension found in all other extant apes (White et al., 2015). It is therefore difficult to explain what selection might underlie a transition from a suspensory/knuckle-walking primate to a palmigrade quadrupedal arboreal climber/clamberer prior to the adoption of terrestrial bipedality. An “adaptively simpler” scenario is that the African ape ancestor shared the primitive morphological state preserved in much of Ar. ramidus. While this scenario requires significant parallelism among African apes, such homoplasy already very likely occurred in gibbons, orangutans and, potentially, other Miocene lineages, such as Morotopithecus, Pierolapithecus, Hispanopithecus (Larson, 1998; MacLatchy, 2004; Moya-Sola & Kohler, 1996; Moya-Sola et al., 2004; Ward, 2015). Moreover, parallelism has been frequently observed during animal evolution, and can be facilitated by shared genetic variation and genomic organization in closely related taxa (Reno, 2014).

It is difficult to overemphasize the special problem in any analysis of hominid evolution: there are no extant models of non-suspensory tailless anthropoids that can be reliably compared to Ar. ramidus, i.e., no living ape is a suitable comparator because all have a long history of substantial vertical climbing and/or suspension. Hominids are unique. ARA-VP-6/500 suggests no history of adaptation to suspension in any of its major anatomical character complexes, including those of the wrist, hand, elbow, humerus, femur, foot, and limb proportions (Lovejoy et al., 2009c; White et al., 2015). Therefore, it very likely retained a sacrum and lumbar column also unmodified for suspension.

In this light, the evidence from atelids raises the possibility that the earliest special adaptation to upright walking in hominids was a similar reduction in upper iliac height, added to a significant degree of exaptive spinal invagination achieved as part of the generally derived hominoid bauplan of the LCA. This is reasonable to presume because moderately broad sacral alae are likely to have already been present. Based on Brachyteles, this would have permitted near or potentially complete simultaneous extension of the hip and knee during erect stance. However, further expansion of the sacral alae was also certainly a possible mechanism for further lumbar emancipation, although it may not have been fully developed until the Australopithecus grade of human evolution (Lovejoy, 2005a).

This conclusion bears on the morphology of the pelvis of Ardipithecus, which likely included both a broad sacrum (implied Lovejoy et al., 2009d) and reduced iliac height (observed) but also with a substantially shortened iliac isthmus (observed). The latter suggests considerable age for upright walking in Ardipithecus, since upper iliac shortening as seen in the atelids (which lack lower iliac shortening) may well have preceded any major modifications of the iliac isthmus. Thus the earliest morphological adaptation to upright walking may well have been a shortening of the upper ilium in convergence with atelids (Figs. 8). This change in both taxa serves to facilitate lordosis, however for hominids the target of selection is bipedal stance while in atelids it is caudal suspension.

What likely followed in hominids was a secondary adaptation, viz, a superoinferior abbreviation of the iliac isthmus, whose primary role was to improve trunk control by the anterior gluteals during single support. Such shortening would have also reduced the height of the trunk’s center of mass, an especially important factor during and immediately following heel strike (Lovejoy, 2005a). Again, this hypothesis will be tested by the future discovery of relevant hominid fossils.

Finally, one point requires stringent reiteration as we close our discussion. The possibility and ease with which bipedality could have been adopted are simply considerations required in reconstructing the transition from arboreal clambering and cautious (i.e., low velocity) climbing to terrestrial upright walking—they are not of themselves an adaptive cause for such behavior. The underlying bases of the adoption of bipedality are still far more likely to be related to social and/or demographic forces that bore directly on fitness, rather than simple “locomotor inertia.” Indeed, the 20-fold increase in terrestriality documented over the past 23 years in northern muriquis has been attributed to a coincidental increase in population size and population density (Tabacow, Mendes & Strier, 2009); this demography-driven expansion of their vertical niche may, in turn, underlie the unexpected increase in their fertility (Strier & Ives, 2012). Similar types of forces are likely to have contributed to the shift to upright walking in hominids (White et al., 2015); that is, no matter how facile the transition to bipedality might have been, the adoption of this kinematically unstable means of locomotion almost certainly introduced a period of locomotor disequilibrium that was unlikely to have had, by itself as a locomotor mechanism, any directly positive effect on fitness.

Supplemental Information

Figure S1 Phylogeny used for the pGLS analysis

Generated from the 10KTrees project (Arnold, Matthews & Nunn, 2010). Scalebar = 10 millionyears.

Click here for additional data file.

Figure S2 KNM-MW 13142-M sacral reconstruction

Part of the left ala from the original cast is obscured by the wax and clay to create a fit with innominate articular surface.

Click here for additional data file.

Figure S3 Muriqui (Brachyteles) lumbar vertebra

Note the position of the transverse processes at the base of the pedicle illustrating partial spinal invagination similar to other atelids (Figs.7). Specimen housed at the Museu Nacional de Rio de Janeiro (National Museum of Brazil). Photo credit: Sérgio L. Mendes.

Click here for additional data file.

Table S1 Means and standard deviations (provided for N ≥ 4) for metric data used in the analysis

Click here for additional data file.

Video S1 Video 1 was filmed on 12 November 2011 at 15:16h, with the nearest observer >1, 469 m from the camera

The muriqui video was excerpted without alteration or enhancement from 30-sec video taken by a motion-triggered camera trap (TIGRINUS model 6.0, supplied with camera model DSC W320). The camera trap was deployed inside the forest at the Reserva Particular do Patrimônio Natural—Feliciano Miguel Abdala (RPPN-FMA) in Caratinga, Minas Gerais, Brazil, with the goal of capturing terrestrial behavior by the muriqui group (Matão group) that has been systematically studied by KBS and her students since 1983. The camera trap was located in the Matão group’s home range where previous ground use had been observed. No humans were present or within a 1 km radius of the muriquis at the times when the videos were taken. In both of these videos, adult males descended to the ground on their own initiative to feed on fallen fruits. Adult males in this group spend more time than other age-sex classes on the ground, but all age-sex classes have been observed in ground use (Tabacow, Mendes & Strier, 2009). Minimum distances between observers and the camera were calculated from GPS records by Marlon Lima.

Click here for additional data file.

Video S2 Video 2 was filmed on 17 September 2011 at 12:09h, with the nearest observer >1,184 m from the camera

The muriqui video was excerpted without alteration or enhancement from 30-sec video taken by a motion-triggered camera trap (TIGRINUS model 6.0, supplied with camera model DSC W320). The camera trap was deployed inside the forest at the Reserva Particular do Patrimônio Natural—Feliciano Miguel Abdala (RPPN-FMA) in Caratinga, Minas Gerais, Brazil, with the goal of capturing terrestrial behavior by the muriqui group (Matão group) that has been systematically studied by KBS and her students since 1983. The camera trap was located in the Matão group’s home range where previous ground use had been observed. No humans were present or within a 1 km radius of the muriquis at the times when the videos were taken. In both of these videos, adult males descended to the ground on their own initiative to feed on fallen fruits. Adult males in this group spend more time than other age-sex classes on the ground, but all age-sex classes have been observed in ground use (Tabacow, Mendes & Strier, 2009). Minimum distances between observers and the camera were calculated from GPS records by Marlon Lima.

Click here for additional data file.

Video S3 Excerpt of video of a spider monkey interacting with tourists in Belize

Note the extension of the hip and knee and the lumbar recurvature producing lordosis achieved in the absence of support with the forelimb or tail. The video in not enhanced but audio is removed from video titled Belize Jungle River Cruise: Spider Monkey Walking Around on Boat uploaded to YouTube.com by BigLivigVideos, on June 25, 2012 and accessed on August 22, 2015. Original video can be found at www.youtube.com/watch?v=2z165uYMKTM.

Click here for additional data file.

We thank Yohannes Haile-Selassie, Curator of Physical Anthropology (CMNH), and Judith Cupasko, Curatorial Associate (MCZ), for access to comparative specimens in their care and Lyman Jellema and Mark Omura for curatorial assistance. The field research was conducted with permission from the Brazilian government and CNPq, and from the Abdalla family for permission to work at the RPPN Feliciano Miguel Abdala. We thank the Sociedade para a Preservacão do Muriqui (Preserve Muriqui) for help with logistics, Dr. Sérgio L. Mendes for long-term support and collaboration, and Daniel Ferraz, Marina Schultz de Cristo, and Fernanda P. Tabacow for permission to use their photos of muriquis. Tim Ryan and Arslan Zaidi provided invaluable statistical advice.

Additional Information and Declarations

Competing Interests

Author Contributions

Field Study Permissions

Data Availability

1 We recognize these taxa at the family level: hominidae, panidae, and gorillidae (White et al., 2009b).

Philip L. Reno is an Academic Editor for PeerJ.

Allison L. Machnicki, Karen B. Strier and Philip L. Reno conceived and designed the experiments, performed the experiments, analyzed the data, wrote the paper, prepared figures and/or tables.

Linda B. Spurlock conceived and designed the experiments, performed the experiments, analyzed the data, prepared figures and/or tables, reviewed drafts of the paper.

C. Owen Lovejoy conceived and designed the experiments, performed the experiments, analyzed the data, wrote the paper.

The following information was supplied relating to field study approvals (i.e., approving body and any reference numbers):

The muriqui field observations led by KBS were strictly observational and adhered to the Animal Behavior Society Guidelines for the Treatment of Animals in Behavioral Research and the International Primatological Society’s Code of Best Practices for Field Primatology. Methods were approved by the University of Wisconsin Animal Care Committee (protocol L00104 through April 2011; then a Wildlife Waiver); the Brazilian government, CNPq, and the administration of the field site provided permission.

The following information was supplied regarding data availability:

Data are provided in the Supplemental Information.

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
