# Peer review of "First steps of bipedality in hominids: evidence from the atelid and proconsulid pelvis"

_PeerJ, doi:10.7717/peerj.1521_

## Round 0.1 · original submission · Major Revisions

Three out of the four referees believe that rather major revisions will be required before this study can be accepted for publication. The single "minor revision" is short on detail, but does urge some substantive changes. I concur to a large degree.

There is a consensus that atelid bipedalism is interesting and could be instructive in models of the evolution of upright walking in the human career, but I agree that there is too little information on atelid performance of bipedalism and too much rehash about Ardipitheus. A brief discussion of Ardipithecus -- currently lacking specimens of lumbar vertebrae -- would perhaps be be better placed at the very end of the ms. with how the authors believe atelids illuminate the locomotion of this basal hominin ("hominid" is decidedly old school).
It seems odd that Brachyteles is featured here in photos and field reports, but is missing from comparative data. (Specimens of this genus are rare but do exist -- e.g., Copenhagen). There isn't much real information about bipedalism in atelids -- this link is a video of a spider monkey walking with a BKBH gait:
https://www.youtube.com/watch?v=8UaidTfvgRQ

There are real issue with data anaysis. The new comparative methods should be utilized (per reviewer 1). Posture needs to be separated clearly from locomotion in all discussions. This reviewer offers a compromise suggestion for data access that I endorse (also see reviewer #3). Reviewers # 2 and #4 raise a variety of issues and contest several claims made in the ms.; these will need to be addressed explicitly in your revision.

Reviewer 1 ·

Basic reporting

In general, the manuscript is written clearly, but there were a few statements that require elaboration, citation or were unclear:

L36: No acknowledgement is made of the studies of Williams et al. and others that have argued for only 5 lumbar vertebrae in Australopithecus and the CLCA (contra these authors).

L38: Add ‘laterally’ to ‘flared iliac crest’.

L54: The authors note that bent-hip-bent-knee (BHBK) gait ‘can be fatiguing’ and that this ‘would presumably discourage its habitual adoption’. I suspect that this is simply the author’s intuition here, but can a reference be added to support the statement about fatigue? Also, contrary to this inference, it is well documented that chimpanzees and some other primates (such as capuchin monkeys) use a BHBK bipedal gait as a normal part of their daily locomotor repitore. So, how nonpermissive can this gait really be for habitual adoption?

Throughout the manuscript, the authors describe the act of climbing a tree in at least three different ways, including “deliberate climbing” (L113), “cautious climbing” (L256), “vertical climbing” (L268), and then later describe Ar. ramidus as an “arboreal climber/clamberer” (L261). What do these modifiers differentiate, exactly? Isn’t all tree climbing deliberate, cautious and vertical?

Fig 4: While the schematics are very useful, it would also be useful to see what some of these anatomical structures actually looked like in the cadaveric specimens. Adding some pictures would help.

Experimental design

The experimental design is straightforward and, at least for the comparative anatomy piece, well described. Some more information on the observational data of the muriquis is required. I have a few additional comments on the statistics and the reporting of the sacrum and ilium measurements.

Validity of the findings

The authors report monitoring ‘bouts of terrestrial behavior’ (L140) in muriquis, but the only contribution from this work are three pictures of muriquis standing bipedally (steadied by a tree, in Figs 1C,D?). Was bipedal standing ever on the ground, not near a tree? It would be interesting to see those images as well. Also, why were these animals adopting these bipedal postures? Were these postures maintained for some time (i.e. did they appear to be stable standing postures)? An additional sentence or two would help ensure the reader that these photographs aren't just fleeting moments in the midst of other locomotor behaviors.

Importantly, do these 'bouts of terrestrial behavior' actually include bipedal walking? If so, can the frequency of this behavior be quantified? Its really not clear when (e.g. just prior to climbing a tree, in the middle of overground travel, etc.) or how often this species used these bipedal postures.

On L165-166, the authors state that ‘ … examination of posture and locomotion in these bouts makes it clear that individuals can readily achieve both a fully erect trunk and a completely extended lower limb’. However, this is not clear to the reader at all. There is no supporting video of animals engaged in bipedal locomotion. If the authors have video that shows these animals using an extended knee (or hip, or trunk) during bipedal walking then it needs to be included in the manuscript. In the absence of these data, revisions to the following three statements are required:

(1) L162: Results section subheading "Lordosis and upright walking in atelines". Change to ‘upright standing’ or something similar, but not walking.

(2) L164-166: "… examination of posture and locomotion in these bouts …”. Delete ‘and locomotion’.

(3) L275-277: "Based on Brachyteles, this would have permitted near or even complete simultaneous extension of the hip and knee during bouts of erect stance and bipedality". Based on the available images, it cannot be argued that atelid lumbar and pelvis anatomy permit extended limb bipedal locomotion, since bipedal locomotion data are not reported. Therefore, delete the last two words: ‘and bipedality’.

L238: “Brachyteles can clearly achieve sufficient lordosis to permit full bipedal posture …” I’d note that what the authors have presented here raises the possibility that this species has a lumbar lordosis. But, there are no supporting measurements of the lordosis or limb posture during standing or, perhaps more importantly, during bipedal walking. As such, this statement is a bit heavy-handed. I think this needs to be revised to reflect the limited data that has actually been provided.

L273: Based on the iliac reduction in atelids, the authors suggest “… the earliest adaptation to upright walking in hominids was likely a reduction in upper iliac height (as has occurred in atelids).” The authors note that further sacral widening was another mechanism, but prefer the iliac blade hypothesis. Why? Sacral widening seems like a more direct means for solving the same problem. An additional sentence or two on this might help clarify the thinking of the authors.

The statistical analysis of Fig. 5 is not of the highest technical standards. The regression equations are based on a mix of inter- and intra-specific data points, rather than species mean values alone. Additionally, a phylogenetic comparative method, such as pGLS, should be used here to calculate the intercepts, slopes and CIs for the samples. Also, the test for a difference between monkeys and apes should be done using a phylogenetic ANCOVA.

It does not appear that the ilium and sacral data shown in Figs 3 and 4 have been adequately “… provided or made available in a disciple-specific repository,” as per the requirements of PeerJ. But the standard here is a bit vague. While a supplement containing the raw measurements (including specimen numbers) is ideal, I think that a Table with the species-specific means for the sacral and ilium measurements (including those from the fossils) would be adequate and consistent with expectations within the field. Also, if there is any sexual dimorphism in these traits, then species-specific means should be reported by sex, and a sex-specific analysis would be warranted.

Additional comments

This is a meaningful contribution to our understanding of fossil hominin locomotion. The anatomical information and comparative osteology are new and should be of interest to other researchers interested in fossil hominin locomotion.

However, the weakest piece of this study is the observational data of atelids. The body of evidence for a lumbar lordosis in atelids amounts to three opportunistic photos shown in Fig 1, and no direct measurements. Additionally, there are no data on trunk and hind limb posture during bipedal walking, which is a significant omission. In the continued absence of these data, I note several areas where statements about bipedalism in atelids should be revised.

Reviewer 2 ·

Basic reporting

No comments

Experimental design

No comments

Validity of the findings

No comments

Additional comments

2015:07:5790:0:0:REVIEW: Earliest bipedality in hominids: evidence from the pelvis and locomotion of atelids.

Thank you for soliciting my review of this manuscript. I have chosen to remain anonymous (against the journal's encouragement) ONLY because I am not comfortable with my review being published alongside the manuscript since I did not prepare it in a manner suitable for publication; rather, as per the norm, I have written what I consider fairly confidential comments to the Editor and authors, which I hope all parties find useful.

In this manuscript, Machnicki, Strier, Reno, Spurlock, and Lovejoy propose to use atelid trunk morphology and observations of some aspects of Brachyteles positional behavior to address the evolution of bipedalism in hominids. The authors find that all atelids possess reduced upper iliac blades, which they argue is a mechanism to free lower lumbar vertebrae and allow lordosis for caudal suspension. They link this morphology to tail-assisted bipedal standing in Brachyteles and show pictures clearly demonstrating a curved lower back and the capability of hindlimb extension in this species. They conclude that the reduced upper ilium allows frees lumbar entrapment and allows the lordosis necessary for hindlimb extension and discuss implications for the evolution of bipedalism in hominids. The authors also dedicate a short section to re-reconstructing the Proconsul nyanzae sacrum, which seems somewhat out of place. Perhaps its inclusion in the Methods section and even as a goal of the manuscript would help. The anatomical contributions here are insightful and certainly worth of publication, and I think convergence studies of this kind can be incredibly informative; however, rather than using this information to interpret morphology in fossil hominoids, the authors rely on previous interpretations of Ardipithecus to guide their interpretation of atelid morphology and implications for hominoid evolution. For example, the authors repeatedly dismiss any role of suspensory behavior in the ancestry of Ardipithecus (and hominids generally), yet the only aspect of locomotor behavior that hominoids and atelines have in common is suspensory behavior. I would have liked to see more about the positional behavior of Brachyteles and other atelids – are they palmigrade quadrupedal arboreal climbers and clamberers like the authors suggest Ardipithecus is? The focus on Ardipithecus is demonstrable by examining how much space in the manuscript is dedicated to Ardi vs. atelids – I find about 4 times as much text dedicated to Ardi as to atelids, presumably the focus of the paper (based on its subtitle). Considering that Ardipithecus does not preserve ANY lumbar vertebrae or sacrum save the distal end, I find it odd to focus so much on Ardipithecus in a manuscript on lumbar entrapment and lordosis. Please find my detailed comments and concerns below.

Line 34: I’m not sure it is well established that bipedalism is “maturely developed” in all species of Australopithecus and particularly “well-advanced” in Adripithecus, especially given that the latter is reported to have a highly abducted hallux that couldn’t achieve toe-off. Can the authors provide multiple perspectives on this issue rather than just their own?

L36: “a lumbar spine containing at least 6 functional segments” is misleading for multiple reasons – 1) There is no evidence for any fossil hominid possessing more than 6 lumbar vertebrae defined using any known method. While it’s true that early hominids seem to have 6 lumbar vertebrae using articular facet orientation as a definition, there are none with 7. 2) The functional definition does not seem to be relevant to a study focused on lordosis and “free” lumbars since the articular facet definition ignores the ribcage (i.e., the first lumbar vertebra of hominids with 6 “functional” lumbars is within the ribcage and therefore not “free”). 3) Moreover, the authors jump back and forth between definitions and are not always clear which one they are using – L36 uses the functional definition, L94 is not specific but must refer to the traditional (rib-based) definition because early fossil hominoids (e.g., Proconsul) had at least 7 functional segments, Figure 4 caption discusses cases of 3, 4, 5, and 7 lumbars but the definition used is again unspecified but must refer to the traditional definition; otherwise, all taxa discussed except gibbons would have more lumbar vertebrae. I know this because I am familiar with the literature and anatomy, but most readers will not be and will be confused at best and misled at worst. The authors do appropriately discuss both definitions at L239, and I suggest they use this approach throughout.

L157: Is it “sacral height” (in text here) or “iliac height” (elsewhere, e.g., L176, and in Fig. 3)?

L186: disk should be disc, as in intervertebral disc.

L213: The Proconsul sacral alae are reconstructed. How was this done? With clay? Did they line up the sacral midline with what’s preserved of the pubic symphysis (which is what Ward et al. appear to do)? Was the acetabulum oriented directly laterally? What about the auricular surface? Can the authors provide more details on how their reconstruction was done? If their results are similar (or 1 mm wider) than Ward et al.’s estimate, then what is the point? Is this a validation of Ward et al.’s reconstruction? Are the authors more confident in their reconstruction than Ward et al.’s? If the authors cannot provide more details or are not more confident in their reconstruction than Ward et al.’s, then what’s the point? If kept, some of this should be in the Methods (and probably Introduction) section as it would be an aim of the paper.

L225: How do the data (on sacral alae reconstruction for P. nyanzae) demonstrate anything about change in the hominoid sacrum with tail loss? Figure 5 shows Proconsul falling near Old World monkeys, but equally close (nearly directly on top of) a triangle, which appears to be either a chimpanzee or orangutan (one Proconsul data point falls directly on top of it). If Proconsul falls equally with OWMs and great apes, how can the authors conclude that the sacrum is monkey-like (and thus that proximal sacral shape change did not accompany tail loss)?

L235: “enervate” means to weaken or fatigue. The authors probably mean “innervate”.

L237: Atelids have craniocaudally tall TPs, but they are generally quite thin dorsoventrally. In this sense, they are quite similar to those of hylobatids, but tend to be longer mediolaterally and orient more cranially, whereas those of hylobatids are blunter and laterally-oriented. Great apes and humans have dorsoventrally thicker TPs. I would call the latter robust and those of atelids tall and long, but I guess it is a matter of word choice.

Figure 5: If the authors include humans in a hominoid regression, their “ape” regression line would no doubt change and would probably go from an isometric relationship between sacral centrum breadth and alar breath to one of positive allometry. What are the implications of negative allometry in monkeys vs. positive allometry in hominoids for interpreting Proconsul, if any? Also, within species, there appears to be little relationship between the two variables in hominoids.

L252: I don’t understand how alar narrowing in the Middle Miocene and retention of that morphology in all great ape lineages except hominids is “decidedly more complex” than parallel reduction in alar breadth in gorillas, chimpanzees, and presumably also in orangutans and hylobatids (and fossil taxa like Oreopithecus). It seems like a minimum of five parallelisms (and any other fossil hominoid not on one of those lineages discovered to have a narrow sacrum must be added to that list) is more complex than hominids re-broadening the sacrum for bipedalism from an ape-like ancestor (hylobatids and orangutans have narrow sacra as well, so this is not just an African ape condition), which requires just two changes.

L257: But we’re not talking about a single lineage here. We’re talking about many lineages, including Proconsul and its contemporaries, middle Miocene hominoids, including but not limited to ancestors of crown hominoids, and the latter lineages, including hominids. This is the same story with numbers of vertebrae – Proconsul had six lumbar vertebrae, which the authors see as a clear pathway to five in hominids, but they ignore (or explain away as the result of parallelism) the living hominoids, each of which is more closely related to hominids than Proconsul, Nacholapithecus, and Pierolapithecus. Oreopithecus is largely ignored even though it is in fact very modern ape-like and perhaps a better model for the evolution of bipedalism than Proconsul. It has five ape-like lumbar vertebrae, whereas Proconsul’s lumbar vertebrae are not very ape-like.

L261: “aboreal” should be “arboreal”.

L263: There is no clade (as far as I am aware) in which the extant taxa tell one story (that humans evolved from an African ape like, certainly a great ape like ancestor) and interpretations of fossils tells a completely different one (like that proposed in this manuscript). Reno’s (2014) examples of anoles and sticklebacks demonstrate the amazing tenacity of natural selection and its result of repeated evolution; however, those cases of parallel evolution (in size and plate/spine reduction, etc.) were obvious from modern phylogenetic studies of extant taxa. Fossils were not discovered and interpreted as evidence to reinterpret the evolutionary history of those anole and stickleback lineages as is being done here.

L273: It is refreshing to get a break from Ardi and read about atelids again. This hypothesis, that atelids can inform hominoid evolution as an analogous example of lordosis and hind limb extension through reduction in upper iliac height, is an interesting and informative one. Have the authors considered that the atelid condition is the result of hindlimb/tail suspension generally? It is clearly not about tail-assisted brachiation or terrestrial bipedal standing since howlers also have reduced upper ilia (Fig. 2). Is the bipedal standing with lordosis and hindlimb extension that the authors observe in Brachyteles an exaptation then? What implications does this have for previous interpretations or Ardipithecus and for hominid evolution generally?

Figure 2 caption: Comparison OF?

Figure 5 caption: remove open parenthesis at the end before the period.

Reviewer 3 ·

Basic reporting

I find figure 4 unrevealing. A lateral view (rather than the AP) of the pelvis and lumbar region would make more sense for understanding lumbar lordosis. This would improve the paper but is not required for it to be understood.

The raw data need to be presented or deposited.

Experimental design

There is no data presented about muriqui bipedalism other than the figure and a statement about rarity. What, exactly, was "observed"?

Figure 4 does not demonstrate "significance" of ligamentous differences. Again where is the data?

Validity of the findings

The findings are interesting.

Additional comments

This is an interesting set of observations that make a nice addition to what is known.

Reviewer 4 ·

Basic reporting

Line 158. Cites Fig 3 regarding “relative sacral height” but Fig. 3 depicts relative iliac height, and there is no figure of relative sacral height.

Atelines and atelids are used interchangeably which is confusing.

Hominid is used rather than the now conventional “hominin”. There is a footnote regarding the choice, but it should include a reference.

Line 54 – the statement that BHBK is fatiguing needs a reference.

Line 235 – innervated, not enervated

Experimental design

no comments (issues on experimental design are subsumed within my comments on validity of the findings)

Validity of the findings

This paper examines the morphology of the ilium and lumbosacral region in atelids as a means to understand the morphological features that permit lumbar lordosis and full hind limb extension (ultimately, for understanding evolution of bipedalism in hominins). Atelids are shown to have broad sacral alae and reduced iliac height and are thus presented as an example of parallel evolution with hominins (and distinct from non human hominoids which have narrow sacra and higher iliac blades).

There are several assumptions in the paper that are not explicitly supported, as well as numerous other weaknesses, as delineated below.

Atelids are described as “frequently engaging in caudal suspension, which requires their lower spines to enter into substantial lordosis similar to that of hominids” (line 125-6). At a minimum, this statement needs a reference regarding frequency of caudal suspension, but the larger issue is that the sagittal bending to which their spine is subjected is not exactly comparable to human lordosis. In humans, lordosis is brought about by dorsal wedging of vertebral bodies, in conjunction with wedging of intervertebral discs. It may certainly be the case that atelids require flexibility in sagittal bending due to their locomotion, and this might be associated with ilio-lumbo-sacral features, but I’m not sure it’s appropriate to equate this with human lordosis.
In addition, Brachyteles is shown in several photos standing with an upright trunk and extended hind limb, and it is noted that Brachyteles can “clearly achieve sufficient lordosis to permit full bipedal posture with completely extended hind limbs” (line 239) . Although the photos are compelling, this information is anecdotal. There is no attempt to quantitatively assess “lordosis” or hind limb extension from the figures, and the photos beg the question of whether any other monkeys (e.g. cercopithecoids) ever stand this uprightly, nor do they address that some strepsirrhines (vertical clingers and leapers) use full hind limb extension (presumably without the iliosacral morphology at issue). Moreover, it is not made clear whether these postures are purely in standing, or whether the curved spine and extended hind limb are maintained if they walk bipedally. Finally, only Brachyteles is shown with these postures, but ironically, the data presented in the paper include every atelid EXCEPT Brachyteles. It is concluded that atelids have morphology that allows lordosis, but it has not been established that Ateles, Alouatta or Lagothrix are capable of these postures.

This paper relies on the assumption that lordosis requires separation of the most caudal lumbar vertebrae from the iliac blades (which is morphologically brought about by a reduction in iliac height relative to the lumbosacral junction and broad sacral alae). I agree that reduced iliac height and broader sacra would separate the lowest lumbar from the ilium, but the main effect of this would be in a mediolateral plane. Since lordosis is in a sagittal plane, I am not convinced that the morphological features highlighted here are necessary for permitting lumbar vertebrae to move into a dorsally extended position. This assumption forms the basis for the entire analysis, so I think it should be more strongly explained or demonstrated. Also, the Fig 2 legend notes that the “deep iliac well” of the langur is not an equivalent condition to that of the African apes. Rather, in African apes vertebral motion is said to be restricted because transverse processes articulate directly with the ilia. So, does that imply that transverse processes do NOT articulate directly with ilia in langurs? If so, I reiterate my point above that simply having a taller ilium (or wider sacrum) may not necessarily “greatly restrict” the motion of the langur’s L7 (Fig 2 legend).
Fig. 5. I appreciate the comparative dissection of the ligaments in this anatomical region but this section would have been more useful if the parameters could be quantified in some way – e.g. “thickness” and orientation are discussed but only subjectively. Also, the legend says that “a thin intertransverse ligament runs from L5 to the ilium” in Alouatta (D), but in the figure, it looks like there are intertransverse ligaments between the lumbar vertebrae as well.

There are functional statements throughout the paper which have not been referenced or empirically substantiated:
Line 237. (re atelids) “Their lumbars have robust TPs to accommodate their massive erector spinae” Fig. 2 is cited but only includes a photo of Alouatta. What is meant by robusticity? Not measured here.
Line 184. “retention of a massive erector spinae in the langur” There should be a citation here.
Fig. 2 legend. “The deep iliac well in which the langur’s L7 is positioned greatly restricts its potential motion.” Not tested directly, e.g. with in-vitro experiments on cadavers.
Fig. 5 legend. “These ligaments likely provide lumbar support, while permitting substantial mobility”, “covers less surface area on the ilium …but serves a similar function”, “the ligamentous tissue would not have been as restrictive as in the langur”, “its motion is highly restricted”. All of these are unsubstantiated functional interpretations.

If the basic morphological/functional issues are not well supported than the application of the results to the fossil record (and its implications for Ardipithecus) remain questionable.

---

## Round 0.2 · Major Revisions

I apologize for the delay in rounding up the reviews of your revised and resubmitted ms. Because of these delays, I did seek an opinion from an expert that was not involved in the first round (#5), and I think this individual makes important observations (e.g., it's not "Proconsul" anymore!). I did eventually chase down 3 other re-reviews, and although there is broad agreement that the revised ms. is substantially improved and that the subject is interesting and important, there are still nontrivial issues that should (and can) be addressed prior to acceptance and publication in PeerJ. One reviewer recommends rejection, 2 recommend major revisions, and one only minor changes. I sincerely hope that the authors will accept these comments as constructive criticism and use them to clarify and sharpen their analyses and inferences. I'm not that concerned about allegiance to the hypothetico-deductive framework, but 2 of the 4 reviewers obviously prefer that approach. I also disagree that ratios cannot be used if allometry exists (per #5); that is simply wrong because the ratios will faithfully record the allometry without assuming anything about them being "size-required" modifications. I look forward to seeing your next (and hopefully final) version of this interesting study.

Reviewer 1 ·

Basic reporting

The authors have submitted an improved manuscript. I think this is a stronger and clearer argument. Some additional clarifications are required on the revisions the authors have made to the manuscript.

L77: The authors support the statement that ‘ape BHBK walking is fatiguing’ with a citation to Crompton et al. (1998). I don’t necessarily disagree that ape bipedalism is probably more fatiguing than human walking, but the data in Crompton et al. don’t address ape BHBK walking fatigue at all.

This is because Crompton et al. measured humans walking with BHBK, not ape bipedal walking, and recent studies have clearly demonstrated that human BHBK walking isn't much like ape bipedal walking in terms of actual mechanics (see Foster et al. 2014 JHE; Demes et al. 2015 AJPA). So, I think conclusions about ape musculoskeletal function that are based on humans trying to walk like apes can be considered unreliable, at best.

L134-145: The authors have scaled back the climbing terminology a bit, and added some definitions to make their thinking clearer. However, the definitions they’ve provided for “deliberate climbing” and “vertical climbing sensu stricto” do not appear to differentiate the act of climbing into a tree. Are these definitions intended to be mutually exclusive?

In my reading, “deliberate climbing” refers to what the animal does once it has reached the ‘arboreal canopy’ while, “vertical climbing” sensu stricto refers ‘substrate ascension’. Using these definitions, couldn’t an ape “vertically climb sensu stricto” into a tree trunk and then “deliberately climb” within the canopy? Additionally, what is the difference between “vertical climbing” sensu strico and sensu lato?


The data provided in Table S1 do not include the fossil measurements. The fossil data need to be included to meet the minimal standards of the PeerJ Data Sharing policy. Please include these.

Experimental design

The experimental design is straightforward and well described. They have added some additional information on the field observations, and included some videos which are all welcomed additions. They have also improved their statistical analyses.

Validity of the findings

L179 '... we also surveyed video resources for further examples of bipedal behavior in ate lids (youtube.com and arkive.com).' Please add some information on the search terms used for your video search, so that they are reproducible.

When I searched ‘youtube.com’ using the phrase ‘spider monkey walking’, the first three videos I found were these:

https://www.youtube.com/watch?v=8UaidTfvgRQ
https://www.youtube.com/watch?v=6UYv8M1fD1A
https://www.youtube.com/watch?v=lATaFdqVwhQ

all of these show bipedal Ateles walking overground with inclined trunks and flexed hind limb postures. These videos are difficult to reconcile with the claims that Ateles uses ‘near or actually fully extended hip and knee postures' during bouts of bipedal locomotion (as stated on L 254).

Note that the screenshots in Fig 4 were taken from the fourth video I found in my search, and show an animal standing and 'walking' more-or-less in circles on a boat, I think. It is unclear to me why the authors are focusing on the video selected for Fig 4, rather than the ones the show Ateles actually walking overground over multiple strides? Please include and provide comment on all the relevant results of your video search.


L 347: “For example, Okada (1985) reports that a spider monkey attains a maximum hip angle of 160 during bipedal walking and an angle of 140 at toe off. Each of these are approximately 20 greater than observed in a chimpanzee and gibbons (Okada, 1985).”

This is a useful reference. However, in looking at Fig. 4 in Okada it appears that the 160 and 140 values are slight overestimates of the degree of hip extension for each species, so I’d suggest ‘about 160’ and ‘about 140’. To me, the max value for spider monkey hip angles looks closer to 150 degrees. Nevertheless, it should be recognized that even at 160 the hip is 20 degrees short of the hip being ‘fully extended’ under a vertical trunk (which occurs at or above 180 degrees, as in the illustrative human cyclogram). As such, even a max 160 angle is still a ‘bent hip’ posture.

In addition, the authors have missed a related, and very relevant paper by Yamazaki (1985) in the same volume as the Okada paper. Yamazaki's study includes exemplar hip, knee and ankle kinematics for the bipedal strides of spider monkeys, gibbons, chimpanzees and humans. Looking at Yamazaki's Fig 9, spider monkeys fall out between chimps and gibbons in their hind limb kinematics among the nonhuman primates, in general. These data also indicate (like Okada) that spider monkeys walk with BHBK limb postures, similar to what is observed in apes and consistent with videos listed above.

There is an interesting argument to be made about the hip being less flexed in Ateles than Pan, but it is also quite clear from the papers of Okada and Yamazaki (as well as the videos above) that spider monkeys still walk with a BHBK gait (a point emphasized by Yamazaki, who often groups Ateles and Pan in the discussion of his results). Given the lower back and pelvis morphology of Atelids, these data do not appear to support the view that 'separation of the most caudal lumbar from the iliac wings permits lordosis necessary for complete hind limb extension' (L78-80). Can the authors reconcile these data with their interpretation of the osteology given in the introduction (e.g. L78-80) and elsewhere? If not, some revisions are required.

L204: The authors provide a great deal of useful information about their work on the Proconsol sacrum reconstruction. Could they also provide a picture of the final reconstruction? They mention casting it, and I assume they still have the cast. Readers might usefully compare it to Fig 2.

Abstract: “Upright walking … requires a lumbar lordosis, a ubiquitous feature in all currently known hominids.” While the authors no doubt believe this to be true, the fossil record does not support this statement. There are many hominins for which lower back morphology is unknown, especially in the earliest taxa, like Sahelanthropus and Orrorin.

Reviewer 2 ·

Basic reporting

Everything looks fine. Regarding the raw data, I found the additional data submitted but the authors don't reference this table in their manuscript.

Experimental design

Ok

Validity of the findings

Everything is ok, although the authors could more explicitly identify their speculation, which is rampant in this manuscript (see 'General Comments for the Author').

Additional comments

This revised manuscript is improved; however, I still find many arguments in it to be odd. For example, the authors insist throughout the manuscript that hominids lack suspensory behavior in their ancestry, yet invoke spinal invagination and upper ilium shortening in atelids as a model for Miocene hominoids like Proconsul and Pierolapithecus. These morphologies have clearly evolved in atelids due to use of the prehensile tail, a suspensory behavior. In fact, one could easily argue that the differences observed between Alouatta and atelines that make the latter taxa more similar to hominoids is due to their combination of tail suspension with forelimb suspension. I absolutely fail to see a need to reject suspension except to defend equally odd arguments denying a suspensory ancestry for Ardipithecus ramidus.

Line 351: "...4-5 ribless AND approximately 6 functional lumbar vertebrae." The use of "or" here (in place of the 'and' I've suggested) unnecessary since atelids have both. It isn't an either/or - both morphologies exist and it is only our historical definitions of what it means to be a lumbar vertebra that promote that idea.

L360: Ref. my above comment on tail suspension + forelimb suspension, the authors state that "these features need not have evolved in the context of forelimb brachiation or suspension"; however, examination of Figure 2 (and my own personal knowledge of atelid trunk morphology) shows that Alouatta is not as derived towards the hominoid/hominin condition as Ateles or Brachyteles. Why would that be other than the latter's adoption of forelimb suspension on top of tail suspension?

L371: The authors argue that it is "decidedly more complex" for sacral breadth to be narrow ancestrally than broad, but I think their argument is flawed for multiple reasons. First, is is decidedly less parsimonious to argue that extant apes (and extant ape-like stem apes) evolved narrow sacra in parallel (see attached PDF) than it is to argue that hominins evolved broadened sacra for a different selective reason - bipedality - than broad sacra initially evolved earlier in primate evolution and has largely been maintained through stabilizing selection. This is only a reversal in the broadest sense of the term, so the argument that "reversal implies fluctuating selective pressures in a single lineage" is weak and probably not relevant. Second, the fossil taxa that the authors invoke to demonstrate extensive parallelism in sacral breadth, Ardipithecus and Sivapithecus, do not preserve sacra (or any vertebrae for that matter) relevant to the question. We do not know what the vertebrae of Sivapithecus look like because we don't have any. What if they look like those of orangutans or hylobatids? What if those of Ardipithecus look like extant African apes. The fact is that we don't know because they haven't been recovered yet, so stating otherwise is speculation at best and misleading at worst.

Figure 3: This plot shows a ratio. I cannot find raw data reported in the manuscript. Following the journal's policy that "All authors are responsible for making materials, code, raw data and associated protocols relevant to the submission available without delay," it seems to me that the authors should published their raw data in the supplementary material or at least outline how it will be disseminated.

Annotated reviews are not available for download in order to protect the identity of reviewers who chose to remain anonymous.

Reviewer 4 ·

Basic reporting

No comments.

Experimental design

See "validity of the findings".

Validity of the findings

There are improvements to the paper, but the authors have not adequately addressed my previous concerns and I find the revisions on the whole to be unsatisfactory. In general, I find the organization and goals of the paper to be hard to follow. Problems that persist are as follows:
Videos were added to show bipedalism in atelids but this evidence remains anecdotal. More to the point, statements about “lordosis” or hip/knee extension in atelids remain purely qualitative.
“Lordosis” is said to characterize the spine of atelids during suspension by the tail, but also in bipedal posture. Thus, the iliosacral anatomy of atelids is treated as informative for understanding the evolution of bipedalism. Yet, the paper does not make clear any distinction between atelids that use bipedal posture and those that don’t. I doubt Alouatta does, but its morphology looks like the other atelids. Might this only be informative regarding effects of spinal extension during use of prehensile tail?
My original comment about whether or not the spinal extension seen in atelids should even be called “lordosis” still stands. Morever, my questions regarding similar postures or levels of hip extension that might be found in other primates was not addressed. Here’s another thought: macaques trained to walk bipedally developed lordosis (Preuschoft et al., 1988). Macaques do not have “invaginated” spines, and based on this paper, I presume macaques, like other cercopithecoids have tall ilia and “restrictive” ligaments. How are they able to develop lordosis?
Preuschoft, H., S. Hayama, and M. M. Günther. "Curvature of the lumbar spine as a consequence of mechanical necessities in Japanese macaques trained for bipedalism." Folia Primatologica 50.1-2 (1988): 42-58.
A photo of Brachyteles’ pelvis was added but no data. Many of the sample species have a sample size of 1. Why wasn’t this specimen measured?
Fig. 6 – legend (previously fig 5) is still full of untested and speculative statements about the function of spinal ligaments, and ligament size or thickness have not been quantified.
In the abstract, LCA is defined as the hominid/panid last common ancestor, is ambiguously referred to as “LCA” throughout the paper, and then shows up in Fig. 8. as the LCA of African apes and humans instead. This leads to confusion…I’m not sure which is the intended definition.
Fig. 8a seems to treat wide sacra as primitive to primates, with narrowed sacra evolving independently in all nonhuman hominoids. But that scenario doesn’t acknowledge the paper’s finding that hominids have wider sacra than monkeys or apes. Wouldn’t that require further widening of the sacrum in the hominid lineage? How does this influence the paper’s statement that “The alternative hypothesis, that alar breadth and lumbar column length were first reduced in mid-Miocene hominoids only to then be re-broadened in bipedal hominids is decidedly more complex than the more modest alternative…” That is, wouldn’t the hominid sacrum have to become wider regardless of whether the sacrum was broad as in monkeys, or narrow as in apes?
There still seems to be too much emphasis on Ardipithecus (and assumptions about its phylogenetic placement) in this analysis.
The paper would have benefited from some clearly stated hypotheses.
Line 194. The position of the sacral promontory in relation to the ischiopubic ramus is mentioned, but this was not part of the analysis.
Line 195. States “For size normalization we used the mediolateral breadth of the first sacral body and acetabular diameter”. But the ratio defined in Fig. 3 is relative to the acetabular diameter only.

The extensive new description of the reconstruction in Proconsul needs a figure.

Is “lordose” a word?

Line 198- shouldn’t centrum breadth be the covariate in the ANCOVA?

Line 265: “Since the LCA obviously lacked a prehensile tail,we may ask whether spinal invagination, which was part of the major shift in bauplan that permitted lateralization of the shoulder, was not also a critical exaptation that would eventually facilitate the adoption of upright walking in a descendant of the LCA? Based on TP position in some Miocene hominoids such as Pierolapithecus and a similar TP location in atelids (that facilitates their caudal suspension), we suggest that there is a strong probability that it was.”
Is the above statement just saying that spinal adaptations for upright posture in hominoids are preadaptive for habitual bipedalism? Don’t we already know that? That is, we already know that the position of the transverse processes in apes and humans is a shared adaptation for upright posture.

Fig. 3. Legend. Do you mean p < 0.01, not >?

Reviewer 5 ·

Basic reporting

This paper is written clearly with introduction and background. Some more citations are warranted in places as noted in the comments to the authors. Comments on figures are also included there. It represents a coherent body of work.

Experimental design

The paper needs a stronger hypotheticodeductive framework. The authors should cite a series of features (height below iliac crest of the lumbosacral joint, alar breadth, etc) that they hypothesize to reflect or facilitate lordosis in hominins,then set up the paper to predict that because atelids lordose, these features should be found in these animals as well, and then proceed in that framework. As it stands, the paper wanders about the discussion.

Validity of the findings

See below

Additional comments

This manuscript is apparently a revised version of one submitted previously, although I did not review the original manuscript. I see the response to initial reviewers, and the authors appear to have largely addressed their concerns. The purpose of this manuscript is to highlight homoplasies of the upper pelvis and sacrum between atelids and hominids that the authors argue relate to a capacity to achieve lumbar lordosis that also appear to have been present in Ardipithecus, and so the authors argue this was likely indication that the hominid ancestor would have been capable of lordosis and that the stiff lower back of extant great apes was independently derived. I think the authors are likely correct on this point, and I like the use of a platyrrhine analogy to test the hypothesis that these morphologies are indeed associated with lordosis in a lineage outside of hominoids. It is appropriate for PeerJ

This said, I think there are changes the authors need to make to strengthen their presentation.

Overall, I think the paper would make a stronger case if the hypotheticodeductive framework were strengthened. If the authors could cite a series of features (height below iliac crest of the lumbosacral joint, alar breadth, etc) that they hypothesize to reflect or facilitate lordosis in hominins, they could then set up the paper by hypothesizing that because atelids lordose, these features should be found in these animals as well, and then proceed in that framework. As it stands, the paper wanders about the discussion.

Smaller points:

The Proconsul nyanzae fossil the authors discuss has been attributed to a new genus name, Ekembo (McNulty et al 2015 JHE), so this should be changed in this paper.

Line 54: Hominid should be defined the first time the word appears
Line 57-58: There are far more citations that should be added here, such as Sanders, Haeusler.
Line 60: comma after ramidus. Also, the authors should clarify here (as they do nicely later on about sacral breadth) it is INFERRED interauricular distance and INFERRED habitual toe-off. Line 70-72: This ‘why’ should be presented as more tentative - ‘perhaps’ not ‘likely’
Line 79: Orangutans can achieve full hindlimb extension (Thorpe et al 2007) without any lordosis at all, and with lumbar entrapment. Here and throughout the paper, the authors should consider the anatomy and behavior of orangutans as they directly impact the argument being made.
Line 86: If there is mounting evidence, there should be a lot of citations here. Also, I do not think the authors need to abbreviate last common ancestor, or transverse process, or old and new world monkeys, in this paper. The words are not that frequent, there is no page limit, and it is easier to read without abbreviations.
Lines 93-4: But Ardi has a powerful grasping hallux, which is likely related to ape-like locomotion? And relatively long curved phalanges?
Line 102: “Moderately advanced” warrants definition and explanation. And how does Oreopithecus compare in this regard?
Line 117: There are more authors by far who have discussed Akembo (Proconsul) being an above-branch quadruped that should be cited here.
Line 125: Kohler’s argument about bipedal characters in the Oreopithecus pelvis was based on more than the one crushed skeleton. She also included more fragmentary but undistorted specimens from Basel. This paper and these fossils should be dealt with here.
Line 129-130: Would the existence of a secondary center of ossification affect bipedal function, or could it simply represent homoplasy of a functionally equivalent character?
Line 138: More elliptical than what?
Line 140: Hylobatids have transverse processes that arise from the body-pedicle junction, as does Hispanopithecus and Morotopithecus. And gibbons and siamangs are pretty capable suspensory/climbing animals, so this ‘intermediate’ position does not equate to some sort of ‘intermediate’ behavior. And some of the Miocene hominoids are likely not much different in size from siamangs. And again, orangutans merit mention here.
Line 159-165: Here would be a great place to outline the morphologies hypothesized to relate to lordosis and how the authors will test these hypotheses. E.g. predicting that atelids do in fact lordosis, and that they exhibit the morphologies.
Line 168 on: The first author observed Brachyteles, but not the taxa used in the morhphological analysis. There are Brachyteles specimens available in the Field Museum, and possibly the AMNH. Still, the authors should carefully justify why this genus is representative of all of the others.
Line 184 section: Why were these taxa chosen? How does Hylobates relate to lordosis or no lordosis? It has a longer and presumably more flexible back than larger hominoids if that is what it is meant to represent. How exactly this choice of taxa tests some of the authors’ predictions should be explained.
Line 191 and paragraph: The authorse should note whether they tested for allometry, because if the relationship between variables is not isometric they cannot use ratios (boxplots) to evaluate the data. And why were two body size surrogates chosen? Why are results from both not reported?
Line 230: This section is really results and discussion, which are intermingled throughout. Separating them would make for a clearer paper, but at least acknowledging both is appropriate. And where are data table(s) with results of the metrics taken? Did I miss it/them? It/they is/are necessary.
Line 257 and paragraph: Is there a test for the association between transverse process position and lordosis? I think humans and great apes are similar in this respect, yet differ dramatically in lumbar posture and mobility. And what should transverse processes have to do with tails? I think I am missing something in this paragraph and the next, perhaps the writing could be clarified. And again, how do orangutans fit in here?
Line 273 and paragraph: Here is where setting up the atelids as a test of their hypothesized form-function relationships would be helpful.
Line 282 and paragraph: This discussion should bring in number of lumbar segments as well, as lordotic posture can be achieved in other ways. Somewhere also the lordosis of bipedally trained macaques should be discussed in this paper as well, especially in this context.
Line 301: It may be that the narrow sacrum of apes itself is not what restricts motion, but reflects a stiffer lumbar spine and reduced erector spinae muscle mass, and so as much reflects decreased mobility rather than causing it? Hylobatids have a relatively tight fit of infants in the birth canal compared with apes, so how they fit in here is worth discussing.
Line 339-340: I think old world monkeys have short bony birth canals. They also tend to have narrower sacra than new world monkeys but similarly short pelves.
Line 342: Hispanopithecus and Morotopithecus also seem to have had “partial invagination” as well.
Line 365: Again, narrow alae themselves may not be the only restrictive morphology, but they may be part of a complex of features.
Line 379-380 and 406: Spinal invagination is found in apes and humans and early hominids, including orangs, so I am a bit confused about the relationship with lordosis. This needs to be clarified.
Line 389: Long curved fingers and grasping toes seem related to ape-like locomotion. However, I agree with the authors that extant great apes almost certainly independently acquired their suspensory specializations. Noting also that hylobatids must have evolved suspension independently, and perhaps Morotopithecus, strengthens arguments of postcranial homoplasy in great apes and hominins as well.
Line 430+: Environmental change toward the end of the Miocene with shrinking forests may have played a part as well and is at least worth mentioning.

Figure 2 needs an accompanying data table. It would also be helpful to just write out the taxon names in the figure so the reader doesn’t have to comb the captions to see who is who.
Figure 3 caption: Make parallel construction – “iliaAC height but acetabulUM diameter.” Pick noun or adjective. Also, since atelids are new world monkeys the non-atelids should be renamed in the figure and captions, perhaps ‘nonsuspensory platyrrhines’ or something, as in Figure 7 also. Also, do the authors have an explanation for the lack of overlap between Alouatta, Ateles and Lagothrix? Which of these displays the same amount of lordosis as Brachyteles if any? The behavioral and skeletal samples need to be aligned in this study for it to make sense.
Figure 5 should have accompanying data and analysis. If transverse process position is linked to lordosis (I’m not convinced it is) this would be worth doing.
Figure 6: including taxon names with pictures would make the figure easier to read, as would grouping taxa according to ones that lordose and those that don’t so the reader can link the anatomy with the function.
Figure 7: Which are the two lowest “new world monkey” points? They seem worth discussing. And that all the hominids fall well above the monkey line also.
Figure 8: This figure is incorrectly drawn for what the authors are arguing. Also, the LCA is placed as a sister taxon to African hominoids not as an ancestor in this drawing, and the apomorphies of it are not found in the African hominoids. I think the authors mean that the node of the LCA has those features, so this cladogram is incorrect and does not say what they are arguing. As drawn, these characters are NOT present in the ancestor of chimps, gorillas and humans. Also, why is there the jag in the bonobo clade? Features of the ancestor at a node need to be drawn on the stem not on the terminal branches. I would also prefer to see scientific names used throughout this figure and the entire manuscript rather than common names. In addition, siamangs have only 4-5 lumbar vertebrae, too, so that character also appears on the hylobatid clade. And on this cladogram, both of them, it would be more parsimonious for Akembo (Proconsul) to have lost invagination as it is found on every clade but that one. If the authors want to do a character analysis, they should do one formally so that their presentation is consistent with the data presented and their argument. And what is the outgroup? What happened to other monkeys? Or fossil apes?

---

## Round 0.3 · accepted · Accept

Your careful attention to reviewer suggestions in this version of your contribution is sincerely appreciated. I hope you will agree that some/many of these comments helped to improve the content and organization of the paper.